# Does access to acute intensive trauma rehabilitation (AITR) programs affect the disposition of brain injury patients?

**Sharfuddin Chowdhury**[1]*, **Luke P. H. Leenen**[2]

**1** Trauma Center, King Saud Medical City, Riyadh, Saudi Arabia, **2** Department of Trauma, University Medical Center Utrecht, Utrecht, Netherlands

* dr_smahmud@yahoo.com

**Data Availability Statement:** All relevant data are within the manuscript and its S1 File.

**Funding:** No funding.

## Abstract

Early incorporation of rehabilitation services for severe traumatic brain injury (TBI) patients is expected to improve outcomes and quality of life. This study aimed to compare the outcomes regarding the discharge destination and length of hospital stay of selected TBI patients before and after launching an acute intensive trauma rehabilitation (AITR) program at King Saud Medical City. It was a retrospective observational before-and-after study of TBI patients who were selected and received AITR between December 2018 and December 2019. Participants' demographics, mechanisms of injury, baseline characteristics, and outcomes were compared with TBI patients who were selected for rehabilitation care in the pre-AITR period between August 2017 and November 2018. A total of 108 and 111 patients were managed before and after the introduction of the AITR program, respectively. In the pre-AITR period, 63 (58.3%) patients were discharged home, compared to 87 (78.4%) patients after AITR (p = 0.001, chi-squared 10.2). The pre-AITR group's time to discharge from hospital was 52.4 (SD 30.4) days, which improved to 38.7 (SD 23.2) days in the AITR (p < 0.001; 95% CI 6.6–20.9) group. The early integration of AITR significantly reduced the percentage of patients referred to another rehabilitation or long-term facility. We also emphasize the importance of physical medicine and rehabilitation (PM&R) specialists as the coordinators of structured, comprehensive, and holistic rehabilitation programs delivered by the multi-professional team working in an interdisciplinary way. The leadership and coordination of the PM&R physicians are likely to be effective, especially for those with severe disabilities after brain injury.

## Introduction

Trauma is a significant public health burden in Saudi Arabia due to the high road traffic crash rate [1]. It has severe, devastating, and often life-threatening consequences. Care for the injured after a crash is exceptionally time sensitive; delays in any stage can make a difference in outcomes. Survivors of significant trauma may experience severe functional impairment and reduced quality of life [2]. Severe injuries are often associated with motor, sensory, and

**Competing interests:** The authors have declared that no competing interests exist.

autonomic impairments, including loss of bladder control and bowel evacuation. These contribute to significant morbidity and impact an individual's ability to move about their home and community and bathe and dress independently [3–5].

Advanced rehabilitation services have become critical for enhancing a patient's functional health status following significant trauma. Early intensive rehabilitation significantly reduces dependency, lessening the need for ongoing community care [6]. The World Health Organization recommends that rehabilitation be available and accessible to anyone who experiences a severe traumatic injury [7]. Acute intensive trauma rehabilitation (AITR) programs following traumatic injuries have also improved functional recovery. However, access is often limited and not available at all hospitals [8]. For example, only 1.5% of patients in Beijing, China, receive this service compared with 50% to 58% in Ontario, Canada [9, 10].

In Saudi Arabia, King Fahad Medical City (KFMC) rehabilitation hospital is the only hospital under the Ministry of Health providing holistic rehabilitation services in the region since 2004 [11]. Due to the limited number of inpatient beds and a huge burden of rehabilitation candidates, the waiting time for patients' acceptance is 6–12 months. Another private rehabilitation center is also available that has limited access due to eligibility and insurance coverage. On the other hand, King Saud Medical City (KSMC) in Riyadh is a major trauma center in the region. It receives the most severely injured polytrauma patients from all over the country. A dedicated trauma unit has managed all polytrauma patients since 2016.

KSMC lacked consultant physiatrists, and the physical medicine and rehabilitation (PM&R) department was severely understaffed for such a major hospital in the region until 2017. With only one PM&R registrar, the focus of services was on outpatient care. There were the physiotherapy, occupational therapy, speech-language pathology, and prosthesis and orthosis departments, which were all inadequately involved with patient care because there was no integration between services. Since the trauma unit's inception in 2016, we have been trying to improve trauma care, especially chronic care, for severely injured brain trauma patients. Since then, we have undertaken different administrative measures, including recruiting PM&R professionals, increasing logistics, and collaborating with the KFMC rehabilitation center. As a part of that collaboration, one visiting consultant physiatrist from KFMC used to visit the KSMC trauma unit once a week to assess the chronic trauma patients and select rehabilitation candidates. He also provided expert opinion for interim hospital care before those patients were transferred to his rehabilitation center. The waiting time for the transfer was long. These chronic patients were receiving essential non-intensive hospital care—such as physiotherapy, occupational therapy, tracheostomy care, and speech-language therapy—in the trauma unit. It was causing bed occupation, increased length of hospital stay, and increased cost. Around this time, two more consultant physiatrists joined KSMC. Subsequently, an integrated multidisciplinary AITR program was implemented at KSMC in December 2018.

This study aimed to compare the outcomes regarding discharge destination and length of hospital stay of selected traumatic brain injury (TBI) patients before and after the launch of the multidisciplinary AITR program.

## Materials and methods

### Setting

KSMC is one of the largest hospitals in Saudi Arabia, with 1,400 inpatient beds. KSMC's emergency department (ED) is the busiest in the kingdom [12]. Three physiatrists oversee the KSMC PM&R department. The department started the AITR program in coordination with physiotherapy, occupational therapy, speech-language pathology, prosthesis and orthosis, and the social work department in December 2018.

## AITR

AITR is an acute in-hospital intensive rehabilitation program for a selected group of severely injured trauma patients who receive at least two to three sessions of different therapies—including physiotherapy, speech-language therapy, occupational therapy, and Botox therapy—for three to four hours each day with breaks in between, five days a week, as decided by the physiatrist [13].

The AITR program was created with specific clinical goals. The multidisciplinary team's training was designed to help people regain function, acquire activities of daily living (ADL) independence, and reintegrate into their homes and communities. The program included physical and sensory-motor training from physiotherapy, functional re-training such as self-care and instrumental ADL from occupational therapy, and psycho-social re-training including social skills from speech therapy [13].

## Assessment and selection of TBI patients for AITR

The PM&R department engages with trauma patients' management at an early stage of their in-hospital courses. A consultant physiatrist does weekly rounds on trauma patients to assess needs and select candidates for AITR. When a patient has fully recovered from an acute head injury, has regained consciousness, and is able to participate in rehabilitation, the physiatrist sets up short-term integrated goals for the patients in collaboration with the treating trauma surgeon and other relevant departments (e.g., physiotherapy, occupational therapy, speech-language pathology, prosthesis and orthosis, and social work) in a multidisciplinary team meeting. Assessment is based on a favorable outcome regarding achievement of independence in daily activities after providing the service. The plans are then revised with patient progress. The persistent vegetative and worse-prognosis patients are recommended for nursing care only. After AITR, if a patient is improved with the achievement of independence in daily activities, they are discharged home with outpatient follow-up at our PM&R department as needed. If the patient needs further rehabilitation to achieve the goals, they are transferred to a long-term rehabilitation facility. The selection criteria or preconditions for AITR are described in Table 1.

## Design

This was a retrospective observational before-and-after study of TBI patients who were referred to PM&R and received AITR between December 01, 2018, and December 31, 2019. The TBI patients who died in the hospital, transferred to another hospital during acute care, or were discharged from the hospital after rapid and good recovery that did not require rehabilitation and remained persistent vegetative were excluded. The data were compared with the pre-integration of inpatient rehabilitation for TBI patients who were assessed and selected for rehabilitation care between August 01, 2017, and November 30, 2018.

Before discharge home, a patient had to be able to: (1) execute self-care activities such as feeding, grooming, dressing, and toileting; (2) move from bed to chair/wheelchair/shower chair independently; (3) securely handle household appliances; and (4) walk with or without support inside the ward [13]. These criteria remained the same in pre-AITR and AITR era.

## Data collection

The data of selected TBI patients who were referred for possible rehabilitation care were collected from the PM&R department and trauma unit records. Then for these selected patients, the data of patient demographics, mechanism of injuries, baseline admission characteristics (on presentation to ED), length of stay, and discharge destination in terms of home or

**Table 1. Patient selection criteria for AITR.**

1. The patient must be medically stable.
   - Medical stability refers to optimizing the patient's physical condition, including diseases or dysfunction of the viscera (e.g., respiratory, cardiovascular, gastrointestinal, urologic, endocrine, and neurological disorders).
   - Criteria:
     I. The patient must be afebrile for 48 hours, may have low-grade temperature if a source has been identified and a treatment plan is in place.
     II. The patient must not require suctioning more frequently than every four hours.
     III. The patient should have a stable cardiac rhythm.
     IV. The patient who requires oxygen must have adequate oxygen saturation on portable oxygen.
     V. The patient must be off from continuous positive airway pressure (CPAP), except for sleep apnea treatment.
     VI. If the patient has a chest tube, it must be stable to gravity for at least 48 hours.
     VII. The patient's medical or surgical workup and treatment must be complete.
     VIII. If a patient has nutritional, pain, or wound issues, they must be manageable and not interfere with the therapies.

2. The patient must meet the criteria of at least two of the three (physiotherapy, occupational therapy, and speech-language therapy) major therapy areas.

3. The patient must have the endurance to tolerate at least three to four hours of therapy over the day.

Source: KSMC policy on Intensive Rehabilitation Joint Program, IPP-KSMC-015-V1

rehabilitation center were extracted from the KSMC trauma registry. Discharge destination and time to discharge from hospital were the primary outcome variables.

## Statistical analysis

The data were analyzed using SPSS 25.0 (IBM SPSS Statistics for Windows, Version 25.0. Armonk, NY: IBM Corp.) and R (RStudio Team 2020). The data were subgrouped into the two periods before and after the implementation of AITR. Demographic, mechanism of injury, and baseline injury characteristics (on presentation to ED) were compared to assess equivalence between the two subgroups. The continuous and normally distributed data (e.g., age, respiratory rate, heart rate, systolic blood pressure, international normalized ratio, base excess, $P^H$, abbreviated injury scale [AIS] head, and length of stay) were summarized using mean (standard deviation [SD]) and compared using Student's t-test. Skewed and ordinal data (e.g., Glasgow Coma Scale) were summarized using the median (inter-quartile range [IQR]) and compared using the nonparametric Mann-Whitney U-test. Count data (e.g., male sex, mechanism, trauma team activation, blood transfusion in ED, injury-severity score, and discharge destination) were summarized using proportions and compared using the nonparametric chi-square test. A p-value of $< 0.05$ was considered significant.

## Ethics statement

The study was approved by King Saud Medical City institutional review board (IRB) with a reference number of H1RI-03-Oct18-02. The IRB committee approved a waiver of the requirement to seek informed consent from the participants for a retrospective review of their data.

## Results

A total of 5,933 trauma patients were included in the KSMC registry between August 2017 and December 2019. Of these, 3,419 (57.6%) patients were admitted pre-AITR era and 2,514 (42.4%) patients were admitted after the introduction of AITR. During the pre-AITR period,

2,021 (59.1%) patients sustained TBI, of which 118 (5.8%) died, 1,737 (86%) were discharged after rapid and good recovery that did not require rehabilitation, and 166 (8.2%) became chronic TBI patients (who required rehabilitation care or were persistent vegetative patients). Among the chronic TBI patients, 108 (65%) were selected for the rehabilitation care, and the remaining persistent vegetative patients were selected for nursing care. On the other hand, during the AITR era, 1,578 (62.8%) patients sustained TBI, of which 105 (6.7%) died, 1,315 (83.3%) were discharged after rapid and good recovery that did not require rehabilitation, and 158 (10%) became chronic TBI patients. Among the chronic TBI patients, 111 (70.3%) were selected for AITR, and the remaining persistent vegetative patients were selected for nursing care (Fig 1).

In the combined group of chronic TBI patients selected for rehabilitation care (n = 219), the demographics were mainly young males (195, 89%) with a mean age of 28.2 (SD 14.2) years. The age distribution is presented in the violin plot (Fig 2).

The commonest mechanism of injury was motor vehicle crashes (192, 87.7%) followed by falls (22, 10%) and assaults (5, 2.3%). Eighty-four (38.4%) patients had trauma team activation (TTA) by the ED, and 27 (12.3%) patients received a blood transfusion in the ED. Ninety-three (42.5%) patients had an injury severity score (ISS) between 16 and 25, followed by 81 (37%) patients less than 16 and 45 (20.5%) patients above 26. The average length of ICU and hospital stay were 18 (SD 10.2) and 45.5 (SD 27.8) days, respectively.

The comparison of selected TBI patients' demographics, mechanisms of injury, baseline (on presentation to ED) characteristics, and outcomes between pre-AITR and AITR are described below (Table 2). There was no significant difference in the AIS for the head between the groups (p = 0.437).

In the pre-intervention period, there were 63 (58.3%) patients discharged to home, compared to 87 (78.4%) after the intervention (p = 0.001; chi-squared 10.2). Time to discharge from hospital pre-intervention was 52.4 (SD 30.4) days, which improved to 38.7 (SD 23.1) days after the introduction of the new AITR program (p < 0.001; 95% CI 6.6–20.9) (Table 2).

The comparison of length of hospital stay between pre-AITR and AITR is presented in the violin plot (Fig 3).

## Discussion

We compared two similar groups (head injury) and an almost equal number of patients with two different rehabilitation strategies—AITR vs. non-intensive or minimal in-hospital rehabilitation (pre-AITR)—in two different periods (pre-AITR: Aug 2017–Nov 2018; AITR: Dec 2018–Dec 2019). This study demonstrated that integrating an intensive rehabilitation program with acute trauma care was associated with a significantly higher proportion of patients being discharged home and after a shorter length of stay in the hospital. As a result, the waiting list of rehabilitation candidates and the load on the only rehabilitation center in Riyadh, KFMC, are decreased.

In our cohort, the heart rate (108.7 bpm vs. 96.9 bpm; p = 0.001) at presentation to ED was significantly higher in the pre-AITR than AITR group. As the systolic blood pressure did not change in both groups (125.9 mm Hg vs. 125.5 mm Hg; p = 0.906), the difference in shock status was not significant. Moreover, the weak difference of blood transfusion requirement (16.7% vs. 8.1%; p = 0.054) in ED supports against significant difference in shock status between the two groups. The international normalized ratio (1.2 vs. 1.1; p = 0.004) was significantly higher in the pre-AITR group. Higher blood transfusion requirements in this group should have corrected the coagulopathy. Moreover, trauma system development in Saudi Arabia is recent, and defects in pre-hospital patient transfer could contribute to the difference in

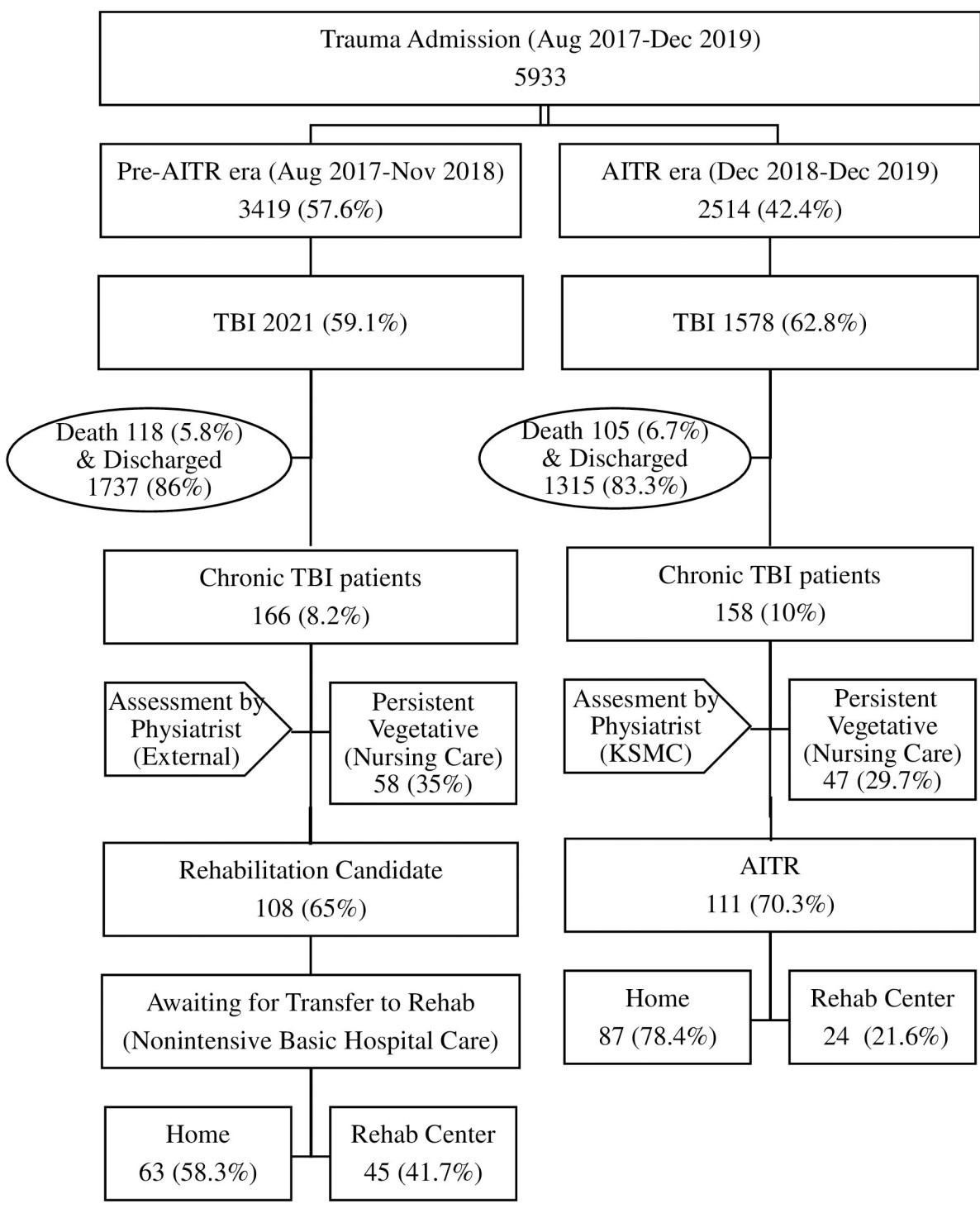

**Fig 1. Sample selection.**

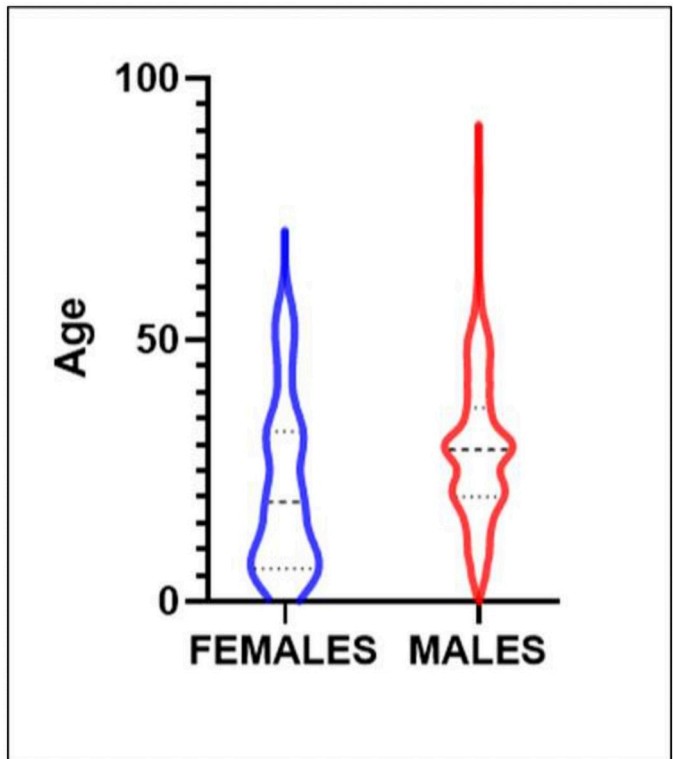

**Fig 2. Age distribution of patients.**

both groups. However, as these parameters are corrected with resuscitation measures immediately in our ED, these should not directly affect the selection of patients or outcomes in both groups.

The TTA rate in the pre-AITR cohort was significantly higher (47.2% vs. 29.7%; p = 0.007). However, ISS was comparatively lower (p = 0.002) in the pre-AITR group, which seems inconsistent as more severe injuries should get more activation. Although TTA is based on specific criteria, it is subjective (at emergency physician discretion) and may have information bias due to over- or under-triage and should not affect outcomes directly.

In our study, ISS in the AITR group was significantly higher (p = 0.002). However, the AIS-head did not show a statistically significant difference (p = 0.437) between the groups, which means we dealt with similar groups of head-injury patients in both rehab strategies. With comparable AIS, the higher ISS in the AITR group indicates more polytrauma [14–16]. On the other hand, with more polytrauma, the AITR group had a significantly lower length of hospital stay (38.7 days vs. 52.4 days; p < 0.001), which favored the success of the AITR program.

Rehabilitation service is an essential pillar of the trauma system and plays a vital role in trauma patients' outcomes. In our situation, it is a challenge to transfer a trauma patient to a rehabilitation center due to a long waiting list. Delay in transfer causes increases in the length of hospital stay and cost. Long waits for rehabilitation have a negative impact on the functional and cognitive recovery of severely injured patients [17].

Identifying factors that contribute to the prediction of discharge disposition is crucial for efficient resource utilization and reducing cost. Several factors may influence discharge location after hospitalization [18]. Functional status due to early and advanced professional

**Table 2. Comparison of selected TBI patients' demographics, mechanisms of injury, baseline (on presentation to ED) characteristics, and outcomes between pre-AITR and AITR.**

| Characteristics | Total (n = 219) | Pre-AITR (n = 108) | AITR (n = 111) | p-value |
|---|---|---|---|---|
| Age (mean years [SD]) | 28.2 (14.2) | 26.9 (14.1) | 29.4 (14.3) | 0.202 |
| Male sex (%) | 195 (89%) | 100 (92.6%) | 95 (85.8%) | 0.097 |
| Mechanism | | | | |
| • Motor Vehicle Collision (%) | 192 (87.7%) | 95 (88%) | 97 (87.4%) | 0.893 |
| • Fall (%) | 22 (10%) | 10 (9.2%) | 12 (10.8%) | 0.694 |
| • Assault (%) | 5 (2.3%) | 3 (2.8%) | 2 (1.8%) | 0.622 |
| Trauma team activation (%) | 84 (38.4%) | 51 (47.2%) | 33 (29.7%) | 0.007* |
| Blood transfusion in ED (%) | 27 (12.3%) | 18 (16.7%) | 9 (8.1%) | 0.054 |
| Respiratory rate (mean breath/min [SD]) | 21.1 (8.2) | 21.4 (8.2) | 20.8 (8.2) | 0.603 |
| Heart rate (mean beat/min [SD]) | 102.7 (25.7) | 108.7 (25.2) | 96.9 (24.9) | 0.001* |
| Systolic blood pressure (mean mm Hg [SD]) | 125.7 (25.3) | 125.9 (28.1) | 125.5 (22.2) | 0.906 |
| Glasgow Coma Scale (median [IQR]) | 7 (5–7) | 7 (5–7) | 7 (4–7) | 0.447 |
| International normalized ratio (mean [SD]) | 1.2 (0.3) | 1.2 (0.3) | 1.1 (0.2) | 0.004* |
| Base excess (mean [SD]) | -3.4 (4.1) | -3.6 (4.3) | -3.2 (3.8) | 0.542 |
| PH (mean [SD]) | 7.32 (0.1) | 7.33 (0.1) | 7.31 (0.1) | 0.204 |
| AIS-head (mean [SD]) | 3.08 (0.76) | 3.05 (0.75) | 3.13 (0.76) | 0.437 |
| Injury severity score (ISS) | | | | 0.002* |
| 1–15 (%) | 81(37%) | 50 (46.3%) | 31 (28.0%) | 0.005* |
| 16–25 (%) | 93 (42.5%) | 42 (38.9%) | 51 (45.9%) | 0.296 |
| > 25 (%) | 45 (20.5%) | 16 (14.8%) | 29 (26.1%) | 0.039* |
| Length of ICU stay (mean days [SD]) | 18 (10.2) | 19.2 (10.8) | 16.9 (9.4) | 0.086 |
| Length of hospital stay (mean days [SD]) | 45.5 (27.8) | 52.4 (30.4) | 38.7 (23.1) | < 0.001* (95% CI 6.6–20.9) |
| Discharge destination | | | | |
| Home (%) | 150 (68.5%) | 63 (58.3%) | 87 (78.4%) | 0.001* |
| Rehab Center (%) | 69 (31.5%) | 45 (41.7%) | 24 (21.6%) | 0.001* |

*Statistically significant at 5% level.

rehabilitation can affect the discharge destination. Nursing home management for all ages has demonstrated a significantly lower quality of life across multiple domains as compared with those living elsewhere [19, 20]. Return to living at home is an important patient-reported outcome following a traumatic injury. Receiving specialized acute rehabilitation is a significant and robust predictor of return to home. Specialized acute intensive rehabilitation helps patients with severe trauma maximize function and independence and return to home. Improving access to specialized acute intensive rehabilitation could potentially reduce discharges to nursing homes or other non-home destinations [21].

The introduction of a multidisciplinary AITR program was a challenge at KSMC. Bringing various allied health-care services under one umbrella was a difficult administrative decision. The PM&R department is understaffed; a dedicated ward with all relevant resources such as a gymnasium is not available to date.

This study is limited to being a retrospective cohort and only a moderate sample of patients. However, it includes consecutive patients during 29 months from the most active trauma center in the country. With only 111 cases of AITR, we attempted to develop a model to improve the trauma patient's outcomes. The study did not analyze in depth the cognitive and functional recovery, bladder and bowel control, etc. An analysis of the Functional Independence Measure (FIM) and the Neurobehavioral Cognitive Status Examination (NCSE) in both groups would

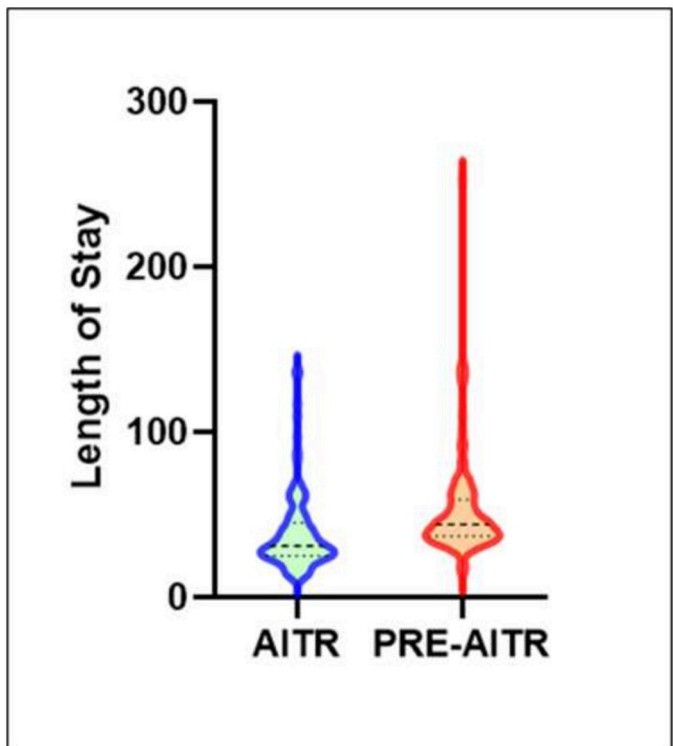

**Fig 3. Comparison of length of hospital stay between pre-AITR and AITR.**

have given more strength to the study. A national trauma registry with a systematic collection of data on patient outcomes would be invaluable to assess such a program.

## Conclusions

The implementation and early integration of AITR significantly reduced the percentage of patients referred to another rehabilitation or long-term facility. It also reduced the length of stay in the hospital. Continuation and expansion of the program to other trauma services with ongoing surveillance are indicated. We also emphasize the importance of PM&R specialists as the coordinators of structured, comprehensive, and holistic rehabilitation programs delivered by the multi-professional team, working in an interdisciplinary way [22]. The leadership and coordination of a PM&R physician are likely to be effective, especially for those with severe disabilities after brain injury.

## Supporting information

**S1 File. Dataset.**
(XLSX)

## Author Contributions

**Conceptualization:** Sharfuddin Chowdhury, Luke P. H. Leenen.

**Data curation:** Sharfuddin Chowdhury.

**Formal analysis:** Sharfuddin Chowdhury.

**Investigation:** Sharfuddin Chowdhury.

**Methodology:** Sharfuddin Chowdhury.

**Project administration:** Sharfuddin Chowdhury.

**Resources:** Sharfuddin Chowdhury.

**Supervision:** Luke P. H. Leenen.

**Validation:** Luke P. H. Leenen.

**Writing – original draft:** Sharfuddin Chowdhury.

**Writing – review & editing:** Sharfuddin Chowdhury, Luke P. H. Leenen.

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
