## [Decision Letter · Decision Letter 0]

30 Jun 2021

PONE-D-21-13676

Does access to acute intensive trauma rehabilitation (AITR) programs affect the disposition of brain injury patients?

PLOS ONE

Dear Dr. asrifine,

Thank you for submitting your manuscript to PLOS ONE. After careful consideration, we feel that it has merit but does not fully meet PLOS ONE’s publication criteria as it currently stands. Therefore, we invite you to submit a revised version of the manuscript that addresses the points raised during the review process.

We look forward to receiving your revised manuscript.

Kind regards,

Angela M. Boutté, Ph.D.

Academic Editor

PLOS ONE

Journal Requirements:

3. Thank you for including your ethics statement: 

"The study was approved by KSMC institutional review board with a reference number of H1RI- 03-Oct18-02. The IRB committee approved a waiver of the requirement to seek informed consent from the participants for a retrospective review of their data.".   

4. Thank you for stating the following in your Competing Interests section:  "no"

 This information should be included in your cover letter; we will change the online submission form on your behalf

5. Please amend your authorship list in your manuscript file to include author andi asrifine

Additional Editor Comments (if provided):

Good day,

The review of your manuscript is complete. A "Minor Revision" per the reviewers commentary is suggested.

Thank you for submitting your work for consideration to PLoS One.

Reviewers' comments:

Reviewer's Responses to Questions

**Comments to the Author**

1. Is the manuscript technically sound, and do the data support the conclusions?

Reviewer #1: Yes

Reviewer #2: Yes

2. Has the statistical analysis been performed appropriately and rigorously? 

Reviewer #1: Yes

Reviewer #2: Yes

3. Have the authors made all data underlying the findings in their manuscript fully available?

Reviewer #1: Yes

Reviewer #2: Yes

4. Is the manuscript presented in an intelligible fashion and written in standard English?

Reviewer #1: Yes

Reviewer #2: Yes

5. Review Comments to the Author

Reviewer #1: Chowdhury and Leenen present the need for early rehabilitation services in the severe traumatic brain injury (TBI) setting and describe the implementation of an acute intensive trauma rehabilitation (AITR) centre in Saudi Arabia at King Saud Medical Centre (KSMC). The AITR addresses understaffing issues and patient waiting times for rehabilitation centre transfer.

The authors present retrospective data before (n=108) and after (n=111) implementation of the AITR. With the introduction of the AITR service, they report a statistically significant increase (p=0.001) in TBI patients discharged home and a reduction in time to discharge from hospital (p=0.0002).

Comments:

1) It is useful to discuss if the discharge home assessment criteria remain the same pre- and post-AITR. How much of the reduced hospital stay is due to earlier, altered, or more proactive assessment versus an improved patient recovery trajectory? I understand such a fine demarcation likely cannot be made in the context of this study, but minimally, it is useful for the reader to understand what has changed or remained static as far as criteria or periodicity of assessment for discharge home.

2) Please discuss the other significant associations on Table 2. In particular, the heart rate and blood transfusion rates both falling in the AITR setting. Is this a product of the selection criteria for the AITR? For instance, the selection criteria around oxygen saturation or stable cardiac rhythm. Is the fall in heart rate in the AITR setting, in part, to do with patient mental wellbeing, or can this be explained by better oxygen saturation in the patients passing the enrolment criteria?

3) Figure 2 needs axes descriptions. Also, the box and whisker plot isn’t very useful at informing the reader around the age distribution demographics, particularly with this distribution skewed in age and gender. Please separate the data by gender and consider a graphic which better presents the distribution, such as a violin or bean plot, or combining the box plot with a beeswarm plot.

Likewise, consider similar graphics presentations for Figure 3.

Reviewer #2: This paper presents a comparison of TBI patient outcomes (specifically, discharge destination and time to discharge) from acute care from before and after implementation of an acute inpatient trauma rehabilitation program. This is an important topic, and this study can support implementation of similar comprehensive inpatient rehabilitation programs in other similar hospital settings. Overall, the paper flows well, with some clarification required (as detailed in comments below).

Abstract

- Line 43: I recommend stating that integration of AITR reduced the percentage of patients referred to another facility (which is what was measured), rather than reducing the need to refer (which is not quite the same). (same for Line 271 under Conclusions in the Discussion)

Introduction

- Lines 72 and 80: consider more neutral language than “suffer from”, such as “experience”, which doesn’t assume a person’s subjective experience the way “suffer from” does.

- Lines 94-97 could be condensed to a single sentence indicating that with only one PM&R registrar, the focus of services was on outpatient care.

- Line 98 ends abruptly (for a long?)

Methods

- Line 131: “assess needs and select candidates for” would be better.

- What do you mean by “assessment is based on a favorable outcome” (line 134)? What specific outcome measures indicate improvement?

- Line 137-138: Does this mean no one receives outpatient therapy outside of a long-term facility?

- What resources are required for the AITR (beyond what was already provided without it)?

- Patient selection criteria #2: what do you mean by “meet the criteria of at least two of the three major therapy areas”? Does this mean that the patient requires at least 2 of 3 or that each of these areas has its own criteria that must be met for a patient to receive their services?

- Line 147-148: I don’t understand what you mean by patients who were discharged from the hospital after completion of treatment being excluded – as written, wouldn’t that be all patients?

- For the primary outcome variable for discharge destination, what were the levels of the variable? (e.g., home, long-term care facility, etc.?)

Results

- What does it mean to become a chronic TBI patient?

- What is “the rehabilitation center” referred to on line 180?

- What would trigger trauma team activation? Since this was more common pre AITR, it is important to note if the need for this intervention indicates differences in the two cohorts that could affect outcomes.

- ISS indicates that the AITR cohort had much more severe polytrauma than the pre-AITR cohort (which was not due to head injury, since AIS were equivalent), yet the pre-AITR cohort had a much higher percentage of trauma team activation (which seems inconsistent?). If the AITR cohort really had more severe polytrauma and still had better outcomes, that is a particularly important point to note. The authors note this briefly in the Discussion (lines 223-225), but don’t expand on the implications or meaning of this.

- Figures are very helpful and clear.

Discussion

- Line 222: be more specific about what you mean by “took off the pressure from”

- Lines 223-240: see comment above about ISS. The points made here about how ISS might affect ICU stay are important, but not really getting at the importance of the ISS difference in this study. More directly stating that, with comparable AIS, the higher ISS in the AITR group indicates more polytrauma, then giving more attention to how this “favors the AITR program’s success” would better highlight the study results.

- The sentence from lines 260-262 is confusing, but seems to make an important point – consider editing for clarity.

6. PLOS authors have the option to publish the peer review history of their article (what does this mean?). If published, this will include your full peer review and any attached files.

Reviewer #1: No

Reviewer #2: No

---

## [Author Response · Author response to Decision Letter 0]

11 Jul 2021

July 11, 2021

Dear Editor,

We want to express our sincere gratitude to the reviewers for their time and effort in reviewing our manuscript. We feel that the comments and recommendations made by the expert reviewers have helped us improve our paper’s quality. Please see our detailed responses to the comments (marked as [reply]) below. Please also refer to the highlighted sentences in yellow for changes that have been made in the manuscript.

Reviewer #1: 

Chowdhury and Leenen present the need for early rehabilitation services in the severe traumatic brain injury (TBI) setting and describe the implementation of an acute intensive trauma rehabilitation (AITR) centre in Saudi Arabia at King Saud Medical Centre (KSMC). The AITR addresses understaffing issues and patient waiting times for rehabilitation centre transfer.

The authors present retrospective data before (n=108) and after (n=111) implementation of the AITR. With the introduction of the AITR service, they report a statistically significant increase (p=0.001) in TBI patients discharged home and a reduction in time to discharge from hospital (p=0.0002).

1) 1st part: It is useful to discuss if the discharge home assessment criteria remain the same pre- and post-AITR. 

[reply] The patients were discharged home when they achieved independence in daily activities. The criteria remained the same in the pre-AITR and AITR era. We rephrased the sentence in the ‘assessment and selection of TBI patients for AITR’ section under ‘materials and methods’ for better clarity as below (page 6, lines 109-112).

After AITR, if a patient is improved with the achievement of independence in daily activities, they are discharged home with outpatient follow-up at our PM&R department as needed. If the patient needs further rehabilitation to achieve the goals, they are transferred to a long-term rehabilitation facility.

We also added the following discharge home criteria (design, page 8, lines 131-134).

Before discharge home, a patient had to be able to: (1) execute self-care activities such as feeding, grooming, dressing, and toileting; (2) move from bed to chair/wheelchair/shower chair independently; (3) securely handle household appliances; and (4) walk with or without support inside the ward. These criteria remained the same in pre-AITR and AITR era. 

1)2nd part: How much of the reduced hospital stay is due to earlier, altered, or more proactive assessment versus an improved patient recovery trajectory? I understand such a fine demarcation likely cannot be made in the context of this study, but minimally, it is useful for the reader to understand what has changed or remained static as far as criteria or periodicity of assessment for discharge home.

[reply] The patient who had a rapid and good recovery that did not require rehabilitation were excluded from the study (design; page 7, line 127-128).

Moreover, the functional recovery was measured by the Functional Independence Measure (FIM) score and available in the patient file. Our study was based on the trauma registry, which does not capture this variable. We want to thank the reviewer as he correctly mentioned to add such fine demarcation likely cannot be made in the context of this study. We will need further IRB approval and further reviews of the patients’ files to add this analysis. We briefly discussed this point in the Discussion as to the limitation of the study (page 15 and lines 259-261).

An analysis of the Functional Independence Measure (FIM) and the Neurobehavioral Cognitive Status Examination (NCSE) in both groups would have given more strength to the study.

2) Please discuss the other significant associations on Table 2. In particular, the heart rate and blood transfusion rates both falling in the AITR setting. Is this a product of the selection criteria for the AITR? For instance, the selection criteria around oxygen saturation or stable cardiac rhythm. Is the fall in heart rate in the AITR setting, in part, to do with patient mental wellbeing, or can this be explained by better oxygen saturation in the patients passing the enrolment criteria?

[reply] Table 2 data are baseline data of patients at first presentation to the emergency department and are corrected with resuscitation measures immediately. So, these variables do not affect the selection of patients. We have added the following paragraph in Discussion (page 13, lines 213-223)

In our cohort, the heart rate (108.7 bpm vs. 96.9 bpm; p = 0.001) at presentation to ED was significantly higher in the pre-AITR than AITR group. As the systolic blood pressure did not change in both groups (125.9 mm Hg vs. 125.5 mm Hg; p = 0.906), the difference in shock status was not significant. Moreover, the weak difference of blood transfusion requirement (16.7% vs. 8.1%; p = 0.054) in ED supports against significant difference in shock status between the two groups. The international normalized ratio (1.2 vs. 1.1; p = 0.004) was significantly higher in the pre-AITR group. Higher blood transfusion requirements in this group should have corrected the coagulopathy. Moreover, trauma system development in Saudi Arabia is recent, and defects in pre-hospital patient transfer could contribute to the difference in both groups. However, as these parameters are corrected with resuscitation measures immediately in our ED, these should not directly affect the selection of patients or outcomes in both groups.

3) Figure 2 needs axes descriptions. Also, the box and whisker plot isn’t very useful at informing the reader around the age distribution demographics, particularly with this distribution skewed in age and gender. Please separate the data by gender and consider a graphic which better presents the distribution, such as a violin or bean plot, or combining the box plot with a beeswarm plot.

Likewise, consider similar graphics presentations for Figure 3.

[reply] Figure 2 and Figure 3 are now updated to violin plot.

Reviewer #2: 

This paper presents a comparison of TBI patient outcomes (specifically, discharge destination and time to discharge) from acute care from before and after implementation of an acute inpatient trauma rehabilitation program. This is an important topic, and this study can support implementation of similar comprehensive inpatient rehabilitation programs in other similar hospital settings. Overall, the paper flows well, with some clarification required (as detailed in comments below).

Abstract

- Line 43: I recommend stating that integration of AITR reduced the percentage of patients referred to another facility (which is what was measured), rather than reducing the need to refer (which is not quite the same). (same for Line 271 under Conclusions in the Discussion)

[reply] Corrected in both places (page 2, line 34; and page 15, lines 265-266). 

Introduction

- Lines 72 and 80: consider more neutral language than “suffer from”, such as “experience”, which doesn’t assume a person’s subjective experience the way “suffer from” does.

[reply] Corrected in both places (page 3, lines 45 and 54).

- Lines 94-97 could be condensed to a single sentence indicating that with only one PM&R registrar, the focus of services was on outpatient care.

[reply] Done (page 4, line 68).

- Line 98 ends abruptly (for a long?)

[reply] Rephrased (page 4, line 70-71).

Methods

- Line 131: “assess needs and select candidates for” would be better.

[reply] Corrected (page 6, line 109).

- What do you mean by “assessment is based on a favorable outcome” (line 134)? What specific outcome measures indicate improvement?

[reply] We added “regarding achievement of independence in daily activities” for clarity (page 6, line 114-115).

A Functional Independence Measure (FIM) and the Neurobehavioral Cognitive Status Examination (NCSE) at selection and after intervention would indicate improvement. These are available in the patient file. Our study is based on the trauma registry, which does not capture this variable. The first reviewer also mentioned this point and stated that such a fine distinction likely could not be made in the context of this study. We will need further IRB approval and further reviews of the patients’ files to add this analysis. We briefly discussed this point in the Discussion as to the limitation of the study (page 15, lines 259-261). 

An analysis of the Functional Independence Measure (FIM) and the Neurobehavioral Cognitive Status Examination (NCSE) in both groups would have given more strength to the study.

- Line 137-138: Does this mean no one receives outpatient therapy outside of a long-term facility? 

[reply] The patients who were discharged home from our facility, our PM&R department followed them in our OPD. Further outpatient therapy as needed were decided by our PM&R team at OPD. But those patients that were transferred to a long-term facility, they were followed up by that respective facility? They do not present to our OPD. For better clarity we rephrased the sentences as below (page 6, lines 116-120).

After AITR, if a patient is improved with the achievement of independence in daily activities, they are discharged home with outpatient follow-up at our PM&R department as needed. If the patient needs further rehabilitation to achieve the goals, they are transferred to a long-term rehabilitation facility.

- What resources are required for the AITR (beyond what was already provided without it)?

[reply] We added the following paragraph to explain AITR better (page 5, lines 100-105). 

 The AITR program was created with specific clinical goals. The multidisciplinary team’s training was designed to help people regain function, acquire activities of daily living (ADL) independence, and reintegrate into their homes and communities. The program included physical and sensory-motor training from physiotherapy, functional re-training such as self-care and instrumental ADL from occupational therapy, and psycho-social re-training including social skills from speech therapy. 

- Patient selection criteria #2: what do you mean by “meet the criteria of at least two of the three major therapy areas”? Does this mean that the patient requires at least 2 of 3 or that each of these areas has its own criteria that must be met for a patient to receive their services?

[reply] It means that the patients require at least 2 of 3 major therapy. We rephrased the sentence in the Table 1 for better clarity (page 7). 

The patient must meet the criteria of at least two of the three (physiotherapy, occupational therapy, and speech-language therapy) major therapy areas.

- Line 147-148: I don’t understand what you mean by patients who were discharged from the hospital after completion of treatment being excluded – as written, wouldn’t that be all patients?

[reply] We rephrased the sentence for better clarity (page 7, lines 126-128). 

The TBI patients who died in the hospital, transferred to another hospital during acute care, or were discharged from the hospital after rapid and good recovery that did not require rehabilitation and remained persistent vegetative were excluded.

- For the primary outcome variable for discharge destination, what were the levels of the variable? (e.g., home, long-term care facility, etc.?)

[reply] For clarity we rephrased in the ‘assessment and selection of TBI patients for AITR’ section (page 6 lines 116-120) as below.

After AITR, if the patient is improved with the achievement of independence in daily activities, they are discharged home with outpatient follow-up at our PM&R department as needed. If the patient needs further rehabilitation to achieve the goal, they are transferred to a long-term rehabilitation facility. 

We also added the criteria for discharge home for both groups (page 8, lines 131-134).

Before discharge home, a patient had to be able to: (1) execute self-care activities such as feeding, grooming, dressing, and toileting; (2) move from bed to chair/wheelchair/shower chair independently; (3) securely handle household appliances; and (4) walk with or without support inside the ward. These criteria remained the same in pre-AITR and AITR era. 

Results

- What does it mean to become a chronic TBI patient?

[reply] Chronic TBI means the patients who require rehabilitation and the patients who are persistent vegetative. Now it is clarified (page 9, lines 165-175) as below.

During the pre-AITR period, 2,021 (59.1%) patients sustained TBI, of which 118 (5.8%) died, 1,737 (86%) were discharged after rapid and good recovery that did not require rehabilitation, and 166 (8.2%) became chronic TBI patients (who required rehabilitation care or were persistent vegetative patients). Among the chronic TBI patients, 108 (65%) were selected for the rehabilitation care, and the remaining persistent vegetative patients were selected for nursing care. On the other hand, during the AITR era, 1,578 (62.8%) patients sustained TBI, of which 105 (6.7%) died, 1,315 (83.3%) were discharged after rapid and good recovery that did not require rehabilitation, and 158 (10%) became chronic TBI patients. Among the chronic TBI patients, 111 (70.3%) were selected for AITR, and the remaining persistent vegetative patients were selected for nursing care (Fig 1). 

- What is “the rehabilitation center” referred to on line 180?

[reply] Here “rehabilitation center” referred to the only rehabilitation center the King Fahad Medical City in Riyadh. We replaced the word with ‘care’ (page 9 line 169).

- What would trigger trauma team activation? Since this was more common pre-AITR, it is important to note if the need for this intervention indicates differences in the two cohorts that could affect outcomes.

[reply] We added an explanation in Discussion (page 13, lines 224-228) as below.

The TTA rate in the pre-AITR cohort was significantly higher (47.2% vs. 29.7%; p = 0.007). However, ISS was comparatively lower (p = 0.002) in the pre-AITR group, which seems inconsistent as more severe injuries should get more activation. Although TTA is based on specific criteria, it is subjective (at emergency physician discretion) and may have information bias due to over- or under-triage and should not affect outcomes directly.

- ISS indicates that the AITR cohort had much more severe polytrauma than the pre-AITR cohort (which was not due to head injury, since AIS were equivalent), yet the pre-AITR cohort had a much higher percentage of trauma team activation (which seems inconsistent?). If the AITR cohort really had more severe polytrauma and still had better outcomes, that is a particularly important point to note. The authors note this briefly in the Discussion (lines 223-225), but don’t expand on the implications or meaning of this.

[reply] Trauma team activation is not a reliable (subjective) variable due to under-or over-triage in terms of patient outcomes, as we stated above. AIS and ISS are reliable as it is based on actual anatomical injury supported by radiological investigations such as pan CT scan. Moreover, all our trauma registry data collectors are AAAM certified and qualified to do so. We rephrased the sentences for clarity in Discussion ass below (page 13-14; lines 231-234).

With comparable AIS, the higher ISS in the AITR group indicates more polytrauma [14, 15]. On the other hand, with more polytrauma, the AITR group had a significantly lower length of hospital stay (38.7 days vs. 52.4 days; p < 0.001), which favored the success of the AITR program.

- Figures are very helpful and clear.

[reply] Thanks so much. We have updated figures 2 and 3 to the violin plot as per the recommendation of the first reviewer.

Discussion

- Line 222: be more specific about what you mean by “took off the pressure from”

[reply] We meant after the introduction of AITR at our hospital more patient is going home instead of waiting to go to a rehabilitation center. So, it decreased the load on the rehabilitation center. We rephrased the sentence as below (Page 13, lines 210-212).

As a result, the waiting list of rehabilitation candidates and the load on the only rehabilitation center in Riyadh, KFMC, are decreased.

- Lines 223-240: see comment above about ISS. The points made here about how ISS might affect ICU stay are important, but not really getting at the importance of the ISS difference in this study. More directly stating that, with comparable AIS, the higher ISS in the AITR group indicates more polytrauma, then giving more attention to how this “favors the AITR program’s success” would better highlight the study results.

[reply] Thanks so much for pointing out this. We deleted the lines 225-239 and added the following sentences (page 13-14; lines 231-234).

With comparable AIS, the higher ISS in the AITR group indicates more polytrauma [14, 15]. On the other hand, with more polytrauma, the AITR group had a significantly lower length of hospital stay (38.7 days vs. 52.4 days; p < 0.001), which favored the success of the AITR program.

- The sentence from lines 260-262 is confusing, but seems to make an important point – consider editing for clarity.

[reply] We have deleted the sentences.

Response to additional editorial comment:

1. Please ensure that your manuscript meets PLOS ONE’s style requirements, including those for file naming. The PLOS ONE style templates can be found at

and

[reply] We updated the revised manuscript accordingly.

2. Please review your reference list to ensure that it is complete and correct. If you have cited papers that have been retracted, please include the rationale for doing so in the manuscript text or remove these references and replace them with relevant current references. Any changes to the reference list should be mentioned in the rebuttal letter that accompanies your revised manuscript. If you need to cite a retracted article, indicate the article’s retracted status in the References list and also include a citation and full reference for the retraction notice.

[reply] The references are now updated as per journal requirements and the reference number 2 is replaced with a new reference:

Stocchetti N, Zanier ER. Chronic impact of traumatic brain injury on outcome and quality of life: a narrative review. Crit Care. 2016 Jun 21;20(1):148. doi: 10.1186/s13054-016-1318-1. PMID: 27323708; PMCID: PMC4915181.

3. Thank you for including your ethics statement: 

“The study was approved by KSMC institutional review board with a reference number of H1RI- 03-Oct18-02. The IRB committee approved a waiver of the requirement to seek informed consent from the participants for a retrospective review of their data.”. 

[reply] KSMC is elaborated to King Saud Medical City.

4. Thank you for stating the following in your Competing Interests section: “no”

Please complete your Competing Interests on the online submission form to state any Competing Interests. If you have no competing interests, please state “The authors have declared that no competing interests exist.”, as detailed online in our guide for authors at http://journals.plos.org/plosone/s/submit-now

 [reply] It is corrected accordingly, and the statement is added in the cover letter. 

5. Please amend your authorship list in your manuscript file to include author andi asrifine

[reply] resolved.

[reply] It is corrected accordingly (page 9, lines 156-160) and the statement is added in the cover letter.

On behalf of all authors,

Yours sincerely,

Dr. Sharfuddin Chowdhury

---

## [Decision Letter · Decision Letter 1]

4 Aug 2021

Does access to acute intensive trauma rehabilitation (AITR) programs affect the disposition of brain injury patients?

PONE-D-21-13676R1

Dear Dr. Chowdhury,

We’re pleased to inform you that your manuscript has been judged scientifically suitable for publication and will be formally accepted for publication once it meets all outstanding technical requirements.

Kind regards,

Angela M. Boutté, Ph.D.

Academic Editor

PLOS ONE

Additional Editor Comments (optional):

Reviewers' comments:

Reviewer's Responses to Questions

**Comments to the Author**

1. If the authors have adequately addressed your comments raised in a previous round of review and you feel that this manuscript is now acceptable for publication, you may indicate that here to bypass the “Comments to the Author” section, enter your conflict of interest statement in the “Confidential to Editor” section, and submit your "Accept" recommendation.

Reviewer #1: All comments have been addressed

Reviewer #2: All comments have been addressed

2. Is the manuscript technically sound, and do the data support the conclusions?

Reviewer #1: Yes

Reviewer #2: Yes

3. Has the statistical analysis been performed appropriately and rigorously? 

Reviewer #1: Yes

Reviewer #2: Yes

4. Have the authors made all data underlying the findings in their manuscript fully available?

Reviewer #1: Yes

Reviewer #2: Yes

5. Is the manuscript presented in an intelligible fashion and written in standard English?

Reviewer #1: Yes

Reviewer #2: Yes

6. Review Comments to the Author

Reviewer #1: (No Response)

Reviewer #2: Thank you for addressing all comments thoroughly and thoughtfully. Though all is understandable and clear, having a scientific editor provide an overall copyediting review would improve the manuscript, as there are several instances of somewhat awkward (if still comprehensible) language.

7. PLOS authors have the option to publish the peer review history of their article (what does this mean?). If published, this will include your full peer review and any attached files.

Reviewer #1: No

Reviewer #2: No

---

## [Editor Report · Acceptance letter]

6 Aug 2021

PONE-D-21-13676R1 

Does access to acute intensive trauma rehabilitation (AITR) programs affect the disposition of brain injury patients? 

Dear Dr. Chowdhury:

I'm pleased to inform you that your manuscript has been deemed suitable for publication in PLOS ONE. Congratulations! Your manuscript is now with our production department. 

Kind regards, 

on behalf of

Dr. Angela M. Boutté 

Academic Editor

PLOS ONE